# Bioinformatics Analysis and Validation of the Role of *Lnc-RAB11B-AS1* in the Development and Prognosis of Hepatocellular Carcinoma

**DOI:** 10.3390/cells11213517

**Published:** 2022-11-06

**Authors:** Dedong Wang, Xiangzhi Hu, Jinbin Chen, Boheng Liang, Lin Zhang, Pengzhe Qin, Di Wu

**Affiliations:** 1Guangzhou Center for Disease Control and Prevention, Guangzhou 510440, China; 2Institute of Public Health, Guangzhou Medical University & Guangzhou Center for Disease Control and Prevention, Guangzhou 510440, China; 3Department of Public Health and Preventive Medicine, School of Medicine, Jinan University, Guangzhou 510632, China; 4Guangzhou Key Laboratory for Clinical Rapid Diagnosis and Early Warning of Infectious Diseases, KingMed School of Laboratory Medicine, Guangzhou Medical University, Guangzhou 510180, China

**Keywords:** hepatocellular carcinoma (HCC), *lnc-RAB11B-AS1*, *RAB11B*, prognosis

## Abstract

*Lnc-RAB11B-AS1* is reported to be dysregulated in several types of cancers and can function as both an oncogene and tumor suppressor gene. To evaluate the potential role of *lnc-RAB11B-AS1* in hepatocellular carcinoma (HCC), we investigated and evaluated its expression in HCC based on the data mining of a series of public databases, including TCGA, GEO, ICGC, HPA, DAVID, cBioPortal, GeneMIANA, TIMER, and ENCORI. The data showed downregulation of *lnc-RAB11B-AS1* in HCC and was accompanied by the synchronous downregulation of the targeted *RAB11B* mRNA and its protein. Low expression of *lnc-RAB11B-AS1* was associated with shorter overall survival (OS) and disease-free survival (DFS) of HCC patients, PD1/PD-L1 was correlated with low expression of *RAB11B*. Furthermore, Gene Ontology (GO) functional annotation and Kyoto Encyclopedia of Genes and Genomes (KEGG) pathway enrichment analysis showed a correlation between immune cell change and non-alcoholic fatty liver disease. The above findings revealed that *lnc-RAB11B-AS1* was down-regulated in HCC and closely associated with the clinical stage of the HCC patients, suggesting that *lnc-RAB11B-AS1* could be a possible predictor for HCC and a potential new therapeutic target for the treatment of HCC.

## 1. Introduction

Hepatocellular carcinoma (HCC) is a common malignant tumor that occurs in the digestive tract with high malignancy, rapid progression, high recurrence rate, and poor prognosis [1]. With approximately 906,000 new cases and 830,000 deaths recorded each year, HCC is the sixth leading cause of cancer incidence worldwide, and the number of associated deaths was ranked third after gastric cancer and esophageal cancer in 2020 [2]. Notably, approximately half of newly diagnosed HCC cases globally occur in China; and as a result, HCC has become the fourth most common malignant tumor and the second most lethal cause of cancer in China, which seriously increases the disease burden [3]. The known risk factors include viral infection, poisonous substances, abnormal metabolism, liver cirrhosis, and smoking [4]. These factors are likely to lead to genetic and epigenetic changes, breaking the internal balance between proto-oncogenes and tumor suppressor genes, and thus resulting in HCC [5]. Research has shown that the sensitivity and specificity of early diagnosis of HCC remain low, and most HCC patients are diagnosed at the middle or even at the late stages. These patients often have severe liver cirrhosis, and it is not suitable for them to undergo surgical resection or liver transplantation. In the past decade, although great progress had been made in non-drug therapies and drug therapies for HCC treatment, such as liver resection, liver transplantation, transcatheter arterial chemoembolization, ablation therapy, and radiotherapy, 50–70% of HCC patients with radical resection may relapse or metastasize within 5 years, which indicates a poor progress for HCC patients [6]. Furthermore, multidrug resistance is one of the predominant reasons for the unsuccessful treatment of HCC because of the reduced sensitivity of patients to chemotherapy drugs. Therefore, exploring the molecular mechanism and seeking new biomarkers to inhibit metastasis and recurrence of HCC is of great importance for the early diagnosis, clinical treatment, and prognosis of HCC.

Long noncoding RNAs (lncRNAs), RNAs without protein-coding functions, are families of noncoding molecular transcripts. In recent years, studies have shown that genomic mutations in non-coding regions are mostly transcribed into lncRNAs and then combined with DNA, RNA, proteins, or other molecules to play a regulatory role in epigenetics. Furthermore, lncRNAs have been identified as having effects on multiple biological processes such as proliferation, apoptosis, invasion, migration, metastasis, and epithelial-mesenchymal transition of multiple cancer cells [7]. Antisense long noncoding RNAs account for approximately 20% of all lncRNAs in mammalian cells and are transcribed from the antisense strands of their coding or non-coding genes. Compared with other types of lncRNAs, such lncRNAs have attracted increasing attention because of their special structure and genomic location [8]. Moreover, accumulating evidence has confirmed that lncRNAs are involved in the progression of HCC, and the underlying mechanism is inextricably related to their subcellular localization. For instance, Yang et al. indicated that cytoplasmic *MAPKAPK5-AS1* acts as a ceRNA by sponging *let-7f-1-3p* and cis-regulating the adjacent gene *MAPKAPK5*, thus promoting the progression of colorectal cancer. While in the nucleus, it can recruit RBM4 and eIF4A1 to promote the translation of *MAPKAPK5* [9].

The RAB protein family belongs to a group of highly conserved small GTP binding proteins in eukaryotes and is an essential member of the RAS superfamily. The activity of RAB proteins is strictly regulated in various cells. They localize specifically in subcellular organelles and have an important role in the vesicular transport of intracellular proteins. More than 70 RAB proteins have been discovered thus far, and they are widely distributed in the membrane structures of various biological cells [10,11]. Studies have documented that different members of the Rab11 subfamily are dysregulated and can modulate the aggressiveness and metastasis of multiple cancers [12], including osteosarcoma [13], lung cancer [14], gastric cancer [15], and renal cell carcinoma [16]. Based on previous research on *lnc-RAB11B-AS1*, it was found that *lnc-RAB11B-AS1* has a regulatory effect on its adjacent gene *RAB11B*. Chen et al. reported that *lnc-RAB11B-AS1* inhibited the proliferation, progression, invasion, and migration of osteosarcoma cells by mediating the effects of its natural antisense transcript *RAB11B* [17]. Li et al. disclosed a novel mechanism by which *lnc-RAB11B-AS1* functions to promote the development of lung cancer through upregulating *RAB11B* expression [13]. However, there is still no relevant explanation of the role of *lnc-RAB11B-AS1* in HCC. Herein, we have investigated the expression level and prognostic value of *lnc-RAB11B-AS1* in HCC by bioinformatic analysis. To elucidate the potential mechanism related to the regulation of *lnc-RAB11B-AS1*, we have examined the interaction between *lnc-RAB11B-AS1* and *RAB11B*. Moreover, an immunohistochemical staining test was conducted to verify our results at the protein level.

## 2. Materials and Methods

### 2.1. Data Acquisition and Preprocessing 

We searched and downloaded the transcriptomic sequencing data and detailed clinical data of HCC patients from TCGA (https://portal.gdc.cancer.gov/repository, accessed on 3 May 2022) database [18]. A total of 369 HCC cases and 50 normal cases were included in our study after duplicate information was deleted. Among 347 HCC patients with known treatment statuses, 239 patients did not receive treatment (4 of them had another new primary tumor), 18 patients were treated, and the remaining 90 cases were recurrent (13 of them had relapses of other tumors, such as lung and bone cancer). Clinicopathological data included age, sex, clinical stage, histological grade, TNM grade, and survival time. Furthermore, two microarray gene expression datasets, GSE144269 and GSE84402, were obtained from GEO (https://www.ncbi.nlm.nih.gov/geo/, accessed on 5 May 2022) database [19]. It appeared that a great number of lncRNAs might be involved in the occurrence of diseases by affecting the expression and regulation of antisense genes [20,21]. Therefore, the correlation between the expression levels of *lnc-RAB11B-AS1* and its protein-coding gene *RAB11B* was explored in HCC.

### 2.2. Immunohistochemistry (IHC) Assay of RAB11B Protein in HCC Tissues

To verify the results in TCGA and GEO, an IHC assay was performed using HCC tissue microarray purchased from Shanghai Outdo Biotech Company (lot NO. hlivh180su15; Shanghai, China,). All experiments followed the manufacturer’s instructions. A total of 90 histologically confirmed HCC cases were enrolled in our study, which contain 90 pairs of cancer tissues and adjacent non-cancerous tissues. The protocol was approved by the ethics committee of Shanghai Outdo Biotech Co., Ltd (The approval code of the ethical committee is SHYJS-CP-1901001). Furthermore, these patients underwent surgery from June 2007 to October 2008 and were followed up for 3–5 years with the recurrence time and multiple biochemical indexes recorded, such as TNM stage, pathological grade, and survival time.

### 2.3. Protein-Protein Interactions of RAB11B

GeneMANIA (http://genemania.org/, accessed on 11 June 2022) [22] can be used to search large and publicly available biological datasets for protein-protein, protein-DNA, and genetic interactions of related genes. The default species “*Homo sapiens*” was selected, and we constructed PPI networks by GeneMANIA online to analyze the interaction between *RAB11B* and other functional proteins.

### 2.4. Human Protein Atlas (HPA)

HPA (https://www.proteinatlas.org, accessed on 4 August 2022) [23] is a public database containing a large-scale human protein map that not only includes tumor tissues but also covers the protein expression of normal tissues. The new section of this database, “single cell type atlas”, includes single-cell RNA sequencing data from 13 human tissues and a total of 192 cell types. It is also available for retrieving genes from different cell types, as well as clustered Uniform Manifold Approximation and Projection plots and expression histograms. On this basis, we explored the expression pattern of *RAB11B* and other marker genes of cluster cells in different single-cell types. To facilitate the comparison, the total read counts for all genes were calculated and normalized to transcripts per million (nTPM) protein-coding genes for each of the single-cell clusters.

### 2.5. Screening of lnc-RAB11B-AS1 Co-Expressed Genes in HCC and Functional Enrichment Analysis

The main features of the DAVID (https://david.ncifcrf.gov, accessed on 11 June 2022) [24] are functional annotation and information links, which can help investigators identify the pathways that are crucial to biological processes. Using GEPIA 2.0 (http://gepia2.cancer-pku.cn/#index, accessed on 11 June 2022) database [25], we searched for the top 200 genes highly correlated with *lnc-RAB11B-AS1*. Next, these co-expressed genes were imported to the DAVID for GO and KEGG enrichment analysis.

### 2.6. GSEA Analysis

To analyze the molecular signal pathways potentially regulated by *lnc-RAB11B-AS1*, GSEA was performed combined with the raw expression profiles from TCGA, as implemented using the “cluserProfiler” R package from the Molecular Signature Database (MSigDB)_v7.0_GMTs and the reference dataset “c5.all.v7.0.entrez.gmt”. Based on a 50% cut-off value, the differentially expressed genes (DEGs) between the *lnc-RAB11B-AS1* high-and low-expression groups were determined at the transcriptomic level. Enrichment results with normalized *p* < 0.05 and a false discovery rate of <0.25 were considered statistically significant.

### 2.7. An Association between lnc-RAB11B-AS1, RAB11B, and Tumor Immune Cell Infiltration

TIMER (https://cistrome.shinyapps.io/timer/, accessed on 18 June 2022) [26] is an integrated database that can be applied to systematically analyze the molecular signatures of tumor-immune interactions, and the levels of six tumor-infiltrating immune subsets can be estimated by the deconvolution method to explore the relationship between the gene expression, mutation, and immune cell infiltrating abundance. Using TIMER, we retrieved a matrix of immune cell infiltration levels in TCGA-LIHC samples, including B cells, CD4^+^ T cells, CD8^+^ T cells, neutrophils, macrophages, and dendritic cells. Correlations between the expression of these genes and various tumor-infiltrating immune cells were investigated using the Spearman test, with *p* <0.05. Furthermore, xCell integrates the advantages of ssGSEA and can calculate the abundance fraction of 64 immune cell types. We assessed the immune cell infiltration scores for each patient in HCC using normalized gene expression profiles. Immune infiltration scores significantly correlated with gene expression were identified using the “corr.test” function of the R package “psych” (version 2.1.6).

### 2.8. Alternation of lnc-RAB11B-AS1 and RAB11B in HCC

The occurrence and development of malignant tumors are influenced by mutations in specific regulatory genes. The cBioportal database (http://www.cbioportal.org, accessed on 3 August 2022) [27] contains multiple databases with a total of 283 large tumor-related genomic and proteomic items and provides visualization tools for the analysis of cancer-related genetic data. Following the conditions of data filtering and extraction, we investigated the type and frequency of *lnc-RAB11B-AS1* and *RAB11B* mutations in HCC.

### 2.9. Correlation between Methylation and lnc-RAB11B-AS1 Expression

Aberrant methylation can be a powerful mechanism leading to the loss of gene function; therefore, we analyzed the relationship between *lnc-RAB11B-AS1* expression and methylation. We downloaded the methylation data, phenotype information, and clinical data of HCC patients from UCSC Xena (https://xenabrowser.net/datapages/, accessed on 1 August 2022) [28], and 50 HCC patients with “normal-tumor” pairings were selected for further analysis.

### 2.10. Drug Susceptibility Analysis

CellMiner (https://discover.nci.nih.gov/cellminer/CellMiner, accessed on 18 July 2022) [29] database facilitates researchers to integrate and analyze the molecular and pharmacological data from the NCI-60 cancer cell line, which is also used by the National Cancer Institute’s Developmental Therapeutics Program to screen more than 100,000 compounds and natural products. We directly selected processed data sets from this database, including RNA-Seq and compound activity (DTP NCI-60). A sensitivity analysis of *lnc-RAB11B-AS1* and related drugs was then conducted and we showed scatter plots of the top nine drugs with the most statistical significance.

### 2.11. Prediction of Potential Candidate miRNAs

We used miRWalk (http://mirwalk.umm.uni-heidelberg.de/, accessed on 8 July 2022) [30] and ENCORI (https://starbase.sysu.edu.cn/, accessed on 8 July 2022) [31] database to predict the miRNAs targeted by *lnc-RAB11B-AS1* and *RAB11B;* thus, the same miRNA binding sites of the two genes could be found. To verify the expression pattern and prognosis of candidate miRNAs, the RNA-Seq data of 232 HCC patients and their corresponding clinical information were retrieved from the International Cancer Genome Consortium (ICGC) (https://dcc.icgc.org/, accessed on 10 July 2022) [32], which collected data on 50 different cancer types including abnormal gene expression, somatic mutations, and epigenetic modifications.

### 2.12. Statistical Analysis

Statistical analyses were performed using GraphPad Prism-8.00 and R-4.1.2. To determine the difference in *lnc-RAB11B-AS1* expression between HCC and normal groups, we used Wilcoxon rank-sum tests via GraphPad Prism. The relationships between *ln-RAB11B-AS1* and clinicopathological features were assessed based on the chi-squared test, which revealed the prognostic significance of *lnc-RAB11B-AS1* for HCC. The Kaplan-Meier method was employed for survival analysis using the log-rank test in group comparisons, while the Cox regression model was utilized to investigate the relationship between clinicopathological features and survival. Hazard Ratio (HR) values and 95% CIs were also calculated. All *p* values were two-tailed, and *p* < 0.05 indicated statistical significance.

## 3. Results

### 3.1. Lnc-RAB11B-AS1 Was Markedly Low Expressed and Positively Correlated with RAB11B in HCC

To explore the expression status of *lnc-RAB11B-AS1* and *RAB11B* in HCC, transcriptome data of the above genes of HCC were retrieved from TCGA. The expression levels of *lnc-RAB11B-AS1* and *RAB11B* were distinctly lower in the single 369 HCC tissues than in the 50 normal tissues (*p* = 0.032, *p* = 0.042, respectively, Figure 1A,D). Analysis of GSE144269 and GSE84402 datasets also verified that *lnc-RAB11B-AS1* and *RAB11B* were down-regulated in HCC tissues compared to normal liver tissues (*p* < 0.001, *p* = 0.025, respectively, Figure 1B,C,E,F). Furthermore, *lnc-RAB11B-AS1* exhibited a positive relationship with *RAB11B* in TCGA-LIHC (R = 0.54, *p* < 0.001, Figure 1G). As depicted in Figure 1H,I, similar association patterns between *lnc-RAB11B-AS1* and *RAB11B* were observed in GSE144269 and GSE84402, but only the former showed statistical significance (R = 0.433, *p* < 0.01; R = 0.521, *p* = 0.059).

### 3.2. The Effects of Overexpressed lnc-RAB11B-AS1 and RAB11B on Clinicopathological Characteristics

As shown in Table 1, there were more men in the high expression group than women (χ^2^ = 9.908, *p* = 0.002). Highly expressed *lnc-RAB11B-AS1* in HCC was associated with a lower level of clinical stage (χ^2^ = 4.054, *p* = 0.044), N stage (χ^2^ = 4.605, *p* = 0.032), and histologic stage (χ^2^ = 4.416, *p* = 0.036). The values of AFP and T-Bil were notably different between *lnc-RAB11B-AS1* high and low groups (*p* = 0.018, *p* = 0.034, respectively). In addition, we found no significant association between *lnc-RAB11B-AS1* expression and age, race, history of hepatocarcinoma risk factors, T stage, M stage, treatment or treatment type, cancer first-degree relative, etc. Therefore, we speculate that high *lnc-RAB11B-AS1* expression indicates a better clinical outcome in HCC patients. As for its sense-cognate gene *RAB11B* shown in Table 2, it was only negatively correlated with the degree of histological grade (χ^2^ = 5.920, *p* = 0.015). In addition, the expression patterns of *lnc-RAB11B-AS1* and *RAB11B* in HCC patients with different clinical stages and histologic grades were visualized through the R package “gg-plot”. The histologic grade and clinical stage tended to be higher in patients with a lower expression of the two genes (Figure 2A–D).

### 3.3. The Relationship between lnc-RAB11B-AS1 Expression and Prognosis of HCC Patients

The Kaplan-Meier analysis suggested that the upregulated expression of *lnc-RAB11B-AS1* was markedly associated with better OS in HCC patients (*p* = 0.022, Figure 2E). OS was evidently reduced in patients with low *RAB11B* expression than in patients with high *RAB11B* expression (*p* = 0.0028, Figure 2F). As an independent validation dataset, the results of *lnc-RAB11B-AS1* and *RAB11B* in GSE144269 uncovered a similar prognostic trend (Figure 2G,H). HCC patients with decreased expression of the above two genes experienced a shorter survival time, but no statistical significance was observed (*p* = 0.12, *p* = 0.21, respectively) due to the deficiency of samples in the GEO data (n = 68). Furthermore, we evaluated the prognostic significance of *RAB11B* for HCC using the Kaplan-Meier Plotter database (https://kmplot.com/analysis/, accessed on 25 May 2022) [33]. As we expected, the prognostic data disclosed OS time, disease-specific survival (DSS) time, progression-free survival (PFS) time, and relapse-free survival (RFS) time was longer in HCC patients with a higher *RAB11B* expression, while those with a low *RAB11B* expression had a worse survival outcome (All *p* < 0.01, Figure 2I).

### 3.4. Cox Regression Model Analysis

As detailed in Table 3, univariate and multivariate Cox regression analyses were employed based on TCGA-LIHC data to investigate the potential risk factors affecting the prognosis of HCC patients. At univariate Cox regression analysis, clinical stage (HR = 1.312; *p* = 0.041), T stage (HR = 2.562; *p* < 0.001), N stage (HR = 1.991; *p* = 0.038), M stage (HR = 3.907; *p* = 0.021), creatinine (HR = 1.710; *p* = 0.031), *lnc-RAB11B-AS1* (HR = 0.814; *p* = 0.009), and *RAB11B* (HR = 0.651; *p* = 0.012) were related to the overall survival rate in HCC patients. The variables with statistical significance in univariate analysis were further included in the multivariate Cox regression analysis, which revealed that poorer overall survival was substantially correlated with a low expression of *lnc-RAB11B-AS1* (HR = 0.799; *p* = 0.025), *RAB11B* (HR = 0.898; *p* = 0.041), advanced clinical stage (HR = 2.628; *p* < 0.001), higher N stage (HR = 8.846; *p* = 0.013), and creatinine (HR = 1.710; *p* = 0.031). According to the above-mentioned data, we have unearthed the potential of *lnc-RAB11B-AS1* as a tumor suppressor gene for HCC.

### 3.5. External Validation of RAB11B Protein Expression and Clinicopathologic Features of HCC Patients

To verify our results at the protein level, the expression of *RAB11B* in 90 HCC tissue chips was detected using immunochemistry, and immunostaining was mainly located in the cytoplasm of HCC cells. As was shown in Figure 3A,B, based on the scoring criteria of cytoplasmic staining, the *RAB11B* expression at the protein level was strikingly decreased in HCC tissues compared with corresponding non-cancerous tissues, which is consistent with the predictive results in public databases. Next, we examined the relationship between *RAB11B* protein expression and clinicopathological characteristics of the 90 HCC patients (Table 4). We found a negative correlation between the *RAB11B* expression and the pathology grade, that is to say, HCC patients with a higher expression of *RAB11B* had a lower pathological grade (χ^2^ = 15.691, *p* < 0.001). In addition, a higher proportion of patients with TB and GGT normal value was observed in the *RAB11B* high expression group compared to that of the low expression group (χ^2^ = 4.107, *p* = 0.043; χ^2^ = 3.903, *p* = 0.048). In the PD-L1 low expression group, there were more HCC patients with an *RAB11B*-high expression (χ^2^ = 9.357, *p* = 0.002), and it showed a distinctly negative correlation between them. However, the remaining clinicopathological factors, including sex, age, clinical stage, TNM stage, recurrence, HBsAg, HBcAb, AntiHCV, ALT, AFP, and CTLA4 did not show any statistical significance. We applied a survival analysis to further assess the prognostic significance of *RAB11B* in HCC, and the results demonstrate that patients with a higher *RAB11B* expression showed longer OS and DFS, suggesting that *RAB11B* might also be a protective factor in the development of HCC (Figure 3C,D).

### 3.6. RAB11B-Associated PPI Network

A total of 20 proteins that have been confirmed or predicted to interact with RAB11B were screened through the GeneMANIA database analysis, including GDI1, TBC1D14, EVI5, TBC1D8B, PCSK9, RAB11FIP1, MYO5B, BICDL1, RAB1B, RAB11A, and so on (Figure 3E). Different colors of nodes represent different interaction relationships, and physical interactions indicate that the corresponding proteins were connected.

### 3.7. RAB11B Expression in Different Cells of HCC

As was shown in Figure 3F, single-cell data from HPA was visualized using UMAP. Each color represents a single cell cluster identified after clustering analysis, and each scatter represents a cell, which has been divided into a total of 16 kinds of cell groups in the liver. The results revealed that the mRNA level of *RAB11B* was higher in T-cells and hepatic stellate cells but was lower in kupffer cells and erythroid cells (Figure 3G). In addition, it disclosed the expression pattern of marker genes in various cell clusters.

### 3.8. Lnc-RAB11B-AS1-Related Genes and Functional Enrichment Analysis

We explored the possible biological function of *lnc-RAB11B-AS1* in the pathogenesis and progression of HCC. The *lnc-RAB11B-AS1* co-expressed genes were submitted to the DAVID for analysis, and a functional annotation table was obtained. As shown in Figure 4A, the results were visualized using a chordal graph with different colors representing the relationships between related genes and various pathways. In GO Biological Processes, the above genes were mainly involved in mitochondrial ATP synthesis-coupled proton transport, aerobic respiration, mitochondrial electron transport, NADH to ubiquinone, hyaluronan metabolic processes, negative regulation of endopeptidase activity, and long-chain fatty acid metabolic processes. In Cellular components, these genes were closely associated with mitochondrial inner membrane, blood microparticles, and extracellular exosomes. Molecular functions were primarily co-regulated with oxidoreductase activity and NADH dehydrogenase (ubiquinone) activity. Additionally, in KEGG analysis, 15 signaling pathways finally showed statistical significance, including in drug metabolism-other enzymes, non-alcoholic fatty liver disease (NAFLD), and oxidative phosphorylation (Figure 4B). The top KEGG enrichment pathway item “NAFLD” is detailed in Figure 4C.

### 3.9. Lnc-RAB11B-AS1-Related Signaling Pathways Obtained by GSEA

HCC patients were divided into two groups according to the expression of *lnc-RAB11B-AS1* and approximately 1721 genes were identified after differential analysis using transcriptional profiles retrieved from TCGA, in which 612 DEGs were upregulated and 1109 were downregulated (adjusted *p* value < 0.05 and |Log2-FC| > 1). The volcano plot depicted in Figure 4D exhibits DEGs, highlighting three up- and downregulated genes with the smallest *p*-values. Then we performed GSEA between *lnc-RAB11B-AS1* high and low groups. The results showed that ATP synthesis-coupled electron transport, axon development, cell adhesion, cell morphogenesis involved in differentiation, and other pathways related to cell-to-cell interactions were significantly enriched (Figure 4E).

### 3.10. Correlation Analysis between lnc-RAB11B-AS1 and RAB11B Expression, and Tumor-Infiltrating Immune Cells in HCC

Elucidating the connection between gene expression and immune infiltration will provide new horizons for HCC immunotherapy in the future. As shown in Figure 5A and 5B, the results revealed that *lnc-RAB11B-AS1* was clearly negatively correlated with six types of immune cells, including B cells (R = 0.23, *p* = 6.1 ×10^−6^), CD8^+^ T cells (R = 0.16, *p* = 2.5 × 10^−3^), CD4^+^ T cells (R = 0.22, *p* = 3.1 ×10^−5^), macrophages (R = 0.35, *p* = 7.3 × 10^−12^), neutrophils (R = 0.36, *p* = 8.8 ×10^−13^) and dendritic cells (R = 0.33, *p* = 1.3 × 10^−10^). Nevertheless, positive relationships were observed between *RAB11B* and several immune cells. Using the xCell algorithm on Sangerbox online website (http://vip.sangerbox.com/home.html, accessed on 15 July 2022), correlations between the expression levels of two genes and multiple immune cell infiltration scores in HCC are shown in Figure 5C. These data further demonstrated that the potential regulatory mechanism of *lnc-RAB11B-AS1* and *RAB11B* in HCC might be brought into effect by influencing immune cell infiltration.

### 3.11. Alternation of lnc-RAB11B-AS1 and RAB11B in HCC

A dataset (TCGA, PanCancer Atlas) containing 372 samples with information on genetic alterations was selected for analysis from the cBioPortal database. The results suggested that the incidence of *RAB11B* mutation was 10% (37/372) while it was only 0.3% in *lnc-RAB11B-AS1* (Figure 5D). The DFS of HCC patients with *RAB11B* gene alteration tended to be lower than that of patients without gene alteration, but it showed a very slight trend toward significance (*p* = 0.286; Figure 5E). In addition, in the tumor subgroup analysis, the expression level of the mutation was associated with the histological grade in HCC (*p* = 0.027; Figure 5F). These findings suggested that the reduction in *RAB11B* mRNA might link to a poor prognosis for HCC but remains to be verified.

### 3.12. Correlation between mRNA Expression and Methylation of lnc-RAB11B-AS1

We downloaded the methylation signal value matrix of HCC patients from UCSC Xena, and finally obtained the data of 14 methylation sites located in the promoter region of *lnc-RAB11B-AS1*. Then, six methylation sites with statistical significance were screened through a comparison between 50 pairs of HCC tissues and normal tissues, including cg08667721, cg02020019, cg02868306, cg04663720, cg12028991, and cg04535340. As shown in Figure 5G, the methylation levels of these methylation regions in the tumor group were significantly higher than those in the normal group. In general, the hypermethylation of the promoter region is negatively correlated with gene expression, and the hypermethylation of the *lnc-RAB11B-AS1* promoter region may lead to the reduction of transcriptional activity, resulting in its decreased expression in HCC. Moreover, we examined the relationship between *lnc-RAB11B-AS1* methylation level and clinical stage in HCC through the “plots” section of the ciBioPortal database (Figure 5H,I). The methylation level was found to be higher in advanced clinical stages, which was consistent with our previous findings that *lnc-RAB11B-AS1* expression was negatively correlated with clinical stage.

### 3.13. Prediction of lnc-RAB11B-AS1 and RAB11B Targeted miRNAs

The Lnclocator database (http://www.csbio.sjtu.edu.cn/bioinf/lncLocator/, accessed on 3 August 2022) [34] was used to predict the subcellular localization of *lnc-RAB11B-AS1* and we found it was mainly localized in the cytosol (Figure 6A). Numerous studies have demonstrated that lncRNAs can affect the post-transcriptional translation process of downstream target genes by interacting with miRNAs. Therefore, we sought to establish a ceRNA network to elucidate the underlying mechanism by which *lnc-RAB11B-AS1* regulates *RAB11B* expression, thus affecting the initiation and development of HCC. Using miRWalk and ENCORI databases, we further performed a prediction of miRNAs targeting *lnc-RAB11B-AS1* and *RAB11B*. The results showed that these two genes had five miRNAs identities in common: *hsa-miR-4640-5p, hsa-miR-3918, hsa-miR-30e-5p, hsa-miR-30a-5p, and hsa-miR-4726-5p*. Because of the limited information on these miRNAs in TCGA and GEO, we continued to perform data mining in the ICGC database. *Hsa-miR-4726-5p* expression in HCC tissues was observed to be strikingly higher than that in normal tissues, while the expression status of other miRNAs was the opposite or was not statistically significant (Figure 6B). The survival curve detailed in Figure 6C revealed that HCC patients with high expression of *hsa-miR-4726-5p* had a worse prognosis, suggesting that it might be a potential hazard factor for HCC. Moreover, we found a negative relationship between *hsa-miR-4726-5p* and *lnc-RAB11B-AS1* based on the ENCORI database (Figure 6D). Furthermore, *hsa-miR-4726-5p* was negatively correlated with *RAB11B* (Figure 6E). To summarize, we made an inference that *lnc-RAB11B-AS1* functions as an antioncogene that positively regulates its adjacent gene *RAB11B* by sponging *hsa-miR-4726-5p* in HCC.

### 3.14. Correlation between lnc-RAB11B-AS1 Expression and Drug Sensitivity

To ensure the reliability of the analysis results, we selected drugs that have been clinically tested and FDA-approved for drug sensitivity analysis. The data revealed high-to-weak positive connections between the expression of *lnc-RAB11B-AS1* and nelarabine, hydroxyurea, pipobroman, thiotepa, chlorambucil, fluphenazine, triethylenemelamine, and uracil mustard. However, the sensitivity to dasatinib gradually increased when *lnc-RAB11B-AS1* expression was elevated (Figure 6F). Considering these findings, *lnc-RAB11B-AS1* might be a novel marker for targeted therapies in HCC.

## 4. Discussion

Thus far, the clinical treatment of HCC has faced a huge bottleneck. Most patients have lost the opportunity for surgical treatment at the time of diagnosis, and chemotherapy remains the main treatment for patients with advanced HCC. With the technological advances in high-throughput sequencing, increasing numbers of functional lncRNAs have been gradually discovered in focused areas [35]. Numerous evidence indicates that lncRNAs are aberrantly expressed in HCC and can be involved in the regulation of initiation, development, and treatment response of HCC. Zhang et al. reported that *lnc-SBF2-AS1* was highly expressed in HCC and could promote HCC metastasis by participating in the regulation of the EMY signaling pathway [36], while Yan et al. confirmed that stress-responsive lncRNA *ROR* knockdown enhanced chemotherapy-induced apoptosis and cytotoxicity [37]. These data all revealed that lncRNAs can not only directly regulate the response of HCC cells to treatment, but also affect the occurrence of HCC by participating in crucial signaling pathways. *Lnc-RAB11B-AS1* is a member of the RAS oncogene family and was found to be dysregulated in several tumors, affecting their biological progress and even resistance to therapeutic agents [38]. As an oncogene, Niu and colleagues demonstrated that *lnc-RAB11B-AS1* stimulated the development and growth of breast cancer cells *in vitro*, as well as tumor angiogenesis and distant metastasis of breast cancer, without affecting primary tumor growth in mice [39]. Li and colleagues found that *lnc-RAB11B-AS1* is overexpressed in lung cancer tissues and is associated with poor prognosis in lung cancer [13]. Feng et al. detected the expression of *lnc-RAB11B-AS1* in 83 gastric cancer tissues and normal tissues using qRT-PCR and found that upregulated *lnc-RAB11B-AS1* in gastric cancer tissues was related to higher stage, lymph node metastasis, distant metastasis, and degree of tumor differentiation. Mechanistically, *lnc-RAB11B-AS1* may inhibit the proliferation, migration and invasion of gastric carcinoma cells by up-regulating *miR-628-3p* and increasing the sensitivity to cisplatin [40]. Meanwhile, the carcinostatic action of *lnc-RAB11B-AS1* was also investigated in colorectal cancer, endometrial cancer, and osteosarcoma by various experiments [17,41]. The association between *lnc-RAB11B-AS1* and HCC progression remains unclear (and relevant studies on it have not yet resolved the matter). More importantly, it has been reported that *lnc-RAB11B-AS1* plays a role in osteosarcoma and lung cancer by regulating the expression of its natural antisense transcript *RAB11B*. Hence, we sought to further explore the regulatory relationship between *lnc-RAB11B-AS1* and *RAB11B* in HCC.

Using TCGA and GEO databases in this study, we found that the expression levels of *lnc-RAB11B-AS1* and its sense-cognate gene *RAB11B* were decreased when compared with paired normal tissues in HCC. Then, an immunohistochemical assay was performed to validate the downregulation of *RAB11B* protein in HCC tissues. A highly positive relationship was observed between the expression of *lnc-RAB11B-AS1* and *RAB11B*. According to a former study, visualization techniques and descriptive indicators can be used to reflect how survival differs with disease subtype, treatment status, and treatment practice patterns [42]. Through screening the clinicopathological information in the database and referring to relevant literature, we first differentiated HCC patients with different treatment statuses in TCGA, among which 249 were untreated, 18 were treated, 90 were recurrent, and 22 were unreported cases. Subsequently, median survival differences between patients with the above three known statuses were assessed. We found that the overall median survival times of untreated, treated, and recurrent HCC patients were 1.43 (IQR 0.05–2.01) years, 1.48 (IQR 0.10–2.43) years, and 2.43 (IQR 1.02–3.48) years, respectively. While in the ICGC database, we only obtained information on whether these HCC patients were treated and the corresponding treatment patterns (177 were untreated and 63 were treated with chemotherapy, surgery or other therapies). We found that the overall median survival of untreated patients was 2.05 years (IQR 1.39–3.21) and that of treated patients was 2.31 years (IQR 1.56–2.88). Patients diagnosed with recurrent HCC experienced longer survival times than untreated HCC patients, consistent with our hypothesis. Unfortunately, for the GSE144269 and GSE84402 dataset, even after reviewing the original text carefully, we have no information on the treatment status of HCC patients.

A correlation analysis of clinicopathological factors revealed that *lnc-RAB11B-AS1* expression was markedly associated with histological grade, clinical stage, AFP, and T-Bil of HCC patients. Our experimental data disclosed a negative correlation between *RAB11B* expression and pathology grade, T-Bil, GGT, and PD-L1. T-Bil and GGT are both important indicators of liver function in patients and are combined with other objective parameters in clinical diagnosis, such as ALBI score, which is considered a gold standard for the diagnosis of HCC patients [43,44]. Abnormally upregulated T-Bil or GGT is often associated with the occurrence of hepatitis, obstructive jaundice, or cirrhosis. Therefore, lower *RAB11B* expression in HCC patients may indicate higher values of the above two indicators, which is closely related to the severity of the disease. Additionally, PD-1/PD-L1 is a hot topic in immunotherapy. PD-L1 is the main ligand of PD-1, which is expressed in various cancer cells and exhibits an inhibitory role in the anti-tumor effect of effector T cells, including in gastric, breast, and liver cancer [45]. A recent study reported that the expression of *lnc-MIAT* was positively correlated with the expression of immune checkpoint molecules in HCC, including PD-1, PD-L1, and CTLA4, influencing the immune microenvironment of HCC [46]. These studies demonstrated the pivotal role of lncRNAs in the tumor microenvironment. In contrast to the functions of the above genes, we found high *RAB11B* expression may lead to the downregulation of PD-L1, thus inhibiting the progression of HCC. The results of the survival analysis showed that the median survival time of patients with low expression of *lnc-RAB11B-AS1* was approximately 45 months, which was less than that identified for patients with high *lnc-RAB11B-AS1* expression, at 70 months. Additionally, elevated *lnc-RAB11B-AS1* was demonstrated to be a protective factor for HCC in the multivariate Cox regression analysis. The above findings suggest that a low expression of *lnc-RAB11B-AS1* indicates an adverse prognosis for HCC and might be a potential molecular marker for prognosis prediction of HCC patients.

Next, we investigated the potential biological function of *lnc-RAB11B-AS1* in HCC. It has been shown to exert a carcinostatic action by affecting multiple signaling pathways. GO and KEGG pathway analyses revealed multiple signaling pathways, such as oxidoreductase activity, drug metabolism-other enzymes, NAFLD, and oxidative phosphorylation. NAFLD is one of the most serious risk factors of HCC, while lncRNAs are reported to play a crucial role in NAFLD. Park JG et al. reported that the degree of hepatic steatosis in NAFLD patients was inversely related to the level of lncRNA *LeXis* in liver tissues [47]. Critical items were identified in GSEA, such as axon development, cell adhesion, and cell morphogenesis. Kong et al. supported the view that the cytoskeletal proteins deregulated by hepatitis B virus X protein (HBx) played a role in cell morphogenesis, adhesion, migration, and proliferation [48]. It is possible that *lnc-RAB11B-AS1* downregulation promotes HCC by similar mechanisms.

Immune-infiltrating cells are an important part of the tumor microenvironment. Studies provided evidence that changes in the number of immune infiltrating cells in tumors can positively or negatively affect the prognosis and immunotherapy efficacy of diseases [49]. The change in *lnc-RAB11B-AS1* expression was found to be negatively correlated with various immune cells, implying that *lnc-RAB11B-AS1* might influence the interaction between immune lymphocytes and malignant tumor cells, and hence regulating the progression of HCC. It appeared that lncRNAs could also participate in the maturation and differentiation of tumor-infiltrating lymphocytes through the regulation of cytokines and participate in the corresponding signaling process of immune regulation, thus affecting the tumor immune function of the organism [50].

DNA sequences can be modified by lncRNAs through changes in chromatin structure, DNA conformation, DNA stability, and gene expression regulation, affecting the occurrence and development of various cancers [51]. Wang et al. also successfully verified that differentially expressed lncRNAs in cancer are modulated by DNA methylation levels using epigenetic data [52]. We further explored the underlying mechanism of *lnc-RAB11B-AS1* downregulation in HCC tissues, and the results indicated that DNA hypermethylation in the promoter region of *lnc-RAB11B-AS1* leads to its reduced expression in HCC. Furthermore, the methylation level increased with HCC clinical stage and histological grade according to the ciBioPortal database, which was consistent with our previous analysis showing the decreased expression of *lnc-RAB11B-AS1* in the advanced stages of HCC.

To our knowledge, lncRNAs regulate signaling pathways and transcription factors through four major modes of action: signal, decoy, guide, and scaffold, thus affecting the biological behavior of cells. The most common mechanism of cytoplasmic lncRNAs is mainly focused on binding miRNAs to act as molecular sponges, interfering with the effects of these miRNAs on target genes. Zhang et al. showed that *KCNQ1OT1*, the overlapping transcript 1 of *lnc-KCNQ1*, acted as a competing endogenous RNA for *miR-506* and promoted PD-L1 expression in HCC [53]. Wang et al. also confirmed that lncRNA *MCM3AP-AS1* promoted the growth of HCC by targeting the *miR-194-5p/FOXA1* axis [54]. According to the prediction of public databases, *lnc-RAB11B-AS1* is mainly located in the cytosol and exhibits a positive regulatory relationship with *RAB11B* in HCC, which is in agreement with the results of previous research showing that *RAB11B* is positively correlated with *lnc-RAB11B-AS1* in lung cancer. Therefore, we attempted to establish an *lnc-RAB11B-AS1-miRNA-RAB11B* axis to test the hypothesis that *lnc-RAB11B-AS1* acts as a ceRNA to regulate downstream mRNA expression. Finally, we have clarified the expression of *hsa-miR-4726-5p* in HCC and its relationships with the two genes and predicted that *lnc-RAB11B-AS1* acts as a ceRNA by sponging *miR-4726-5p* to prevent the progression of HCC via up-regulating *RAB11B* mRNA. Further experiments are planned for the next step. Moreover, individual characteristics and treatment can be combined to achieve better therapeutic effects. We established a direct link between *lnc-RAB11B-AS1* expression and cancer cell sensitivity to chemotherapy, indicating that low *lnc-RAB11B-AS1* expression might be sensitive to multiple chemotherapeutic drugs, such as chlorambucil, hydroxyurea, and thiotepa.

## 5. Conclusions

In conclusion, this study revealed that low expression of *lnc-RAB11B-AS1* was a factor affecting the prognosis of HCC patients. *Lnc-RAB11B-AS1* has also shown promise as a potential molecular marker in the prevention and treatment of HCC.

## Figures and Tables

**Figure 1 cells-11-03517-f001:**
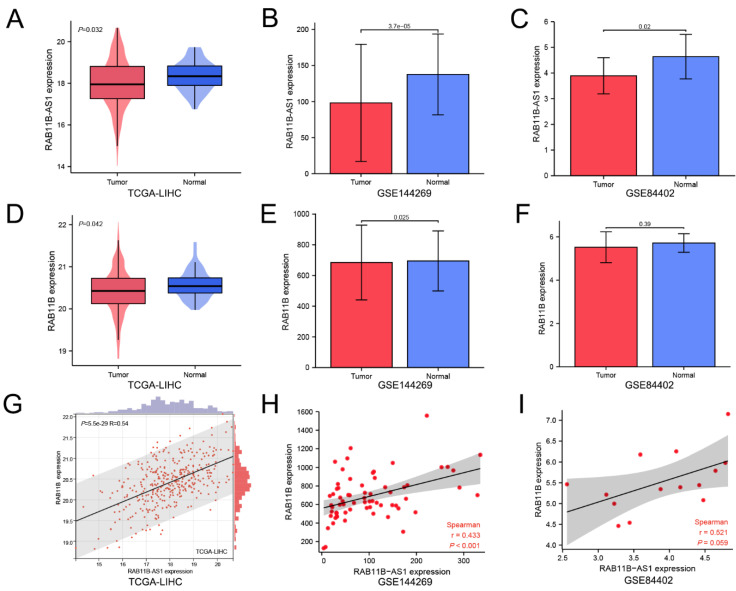
Comparison of *lnc-RAB11B-AS1* and *RAB11B* expression in HCC and normal liver tissues (**A**–**C**) The expression of *lnc-RAB11B-AS1* in HCC tissues and normal tissues based on TCGA-LIHC, GSE144269, GSE84402. (**D**–**F**) *RAB11B* expression in HCC tissues and normal tissues based on TCGA-LIHC, GSE144269, GSE84402. (**G**–**I**) The correlation analysis between *lnc-RAB11B-AS1* and *RAB11B* in three different datasets.

**Figure 2 cells-11-03517-f002:**
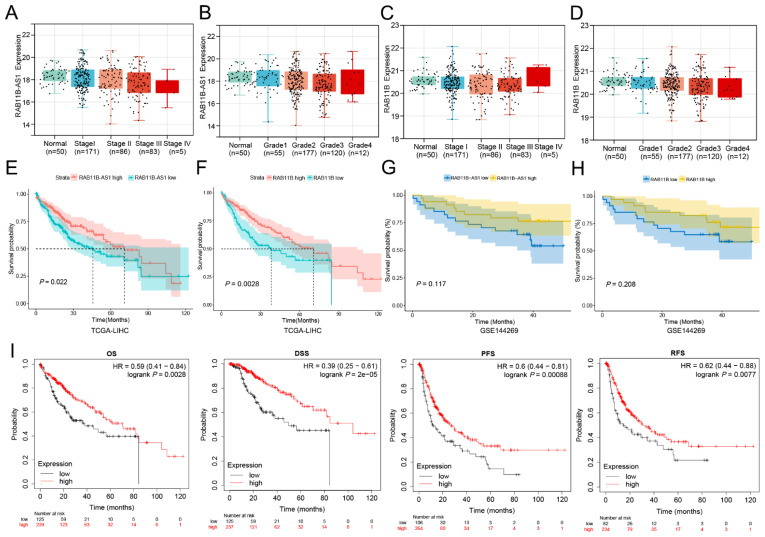
Relationship between *lnc-RAB11B-AS1* and *RAB11B* expression and the prognosis of HCC patients (**A**,**B**) The different expression status of *lnc-RAB11B-AS1* in the histologic grade and clinical stage of HCC. (**C**,**D**) The change of *RAB11B* expression in the histologic grade and clinical stage of HCC. (**E**,**F**) The association between *lnc-RAB11B-AS1* and *RAB11B* expression and OS in TCGA database. (**G**,**H**) The correlation between the expression of *lnc-RAB11B-AS1* and *RAB11B* and OS in GSE144269 dataset. (**I**) The relationships between *RAB11B* expression and OS, RFS, PFS, and DSS of HCC patients in Kaplan-Meier Plotter database.

**Figure 3 cells-11-03517-f003:**
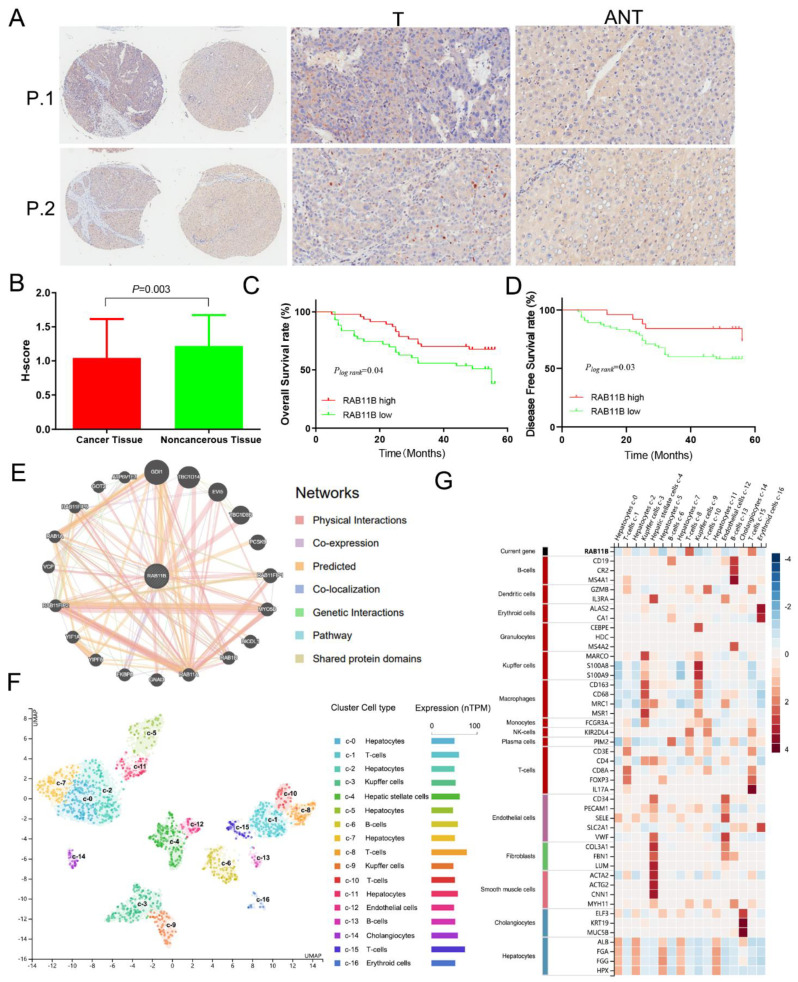
Immunohistochemical analysis of *RAB11B* protein in HCC tissues and adjacent noncancerous tissues. (**A**) Expression of *RAB11B* protein in two representative pairs of HCC tissues (T) and adjacent non-cancerous tissues (ANT), with magnification ×100. (**B**) The score histogram of *RAB11B* expression at protein level in HCC and non-cancerous tissues. (**C**,**D**) The relationship between *RAB11B* expression, OS, and DFS in HCC patients. (**E**) Protein network diagram of interaction with *RAB11B* protein. (**F**) Single-cell UMAP dimensionality reduction map of the liver. (**G**) RNA levels of *RAB11B* and mark genes in different single cell type clusters of the liver (Human Protein Atlas).

**Figure 4 cells-11-03517-f004:**
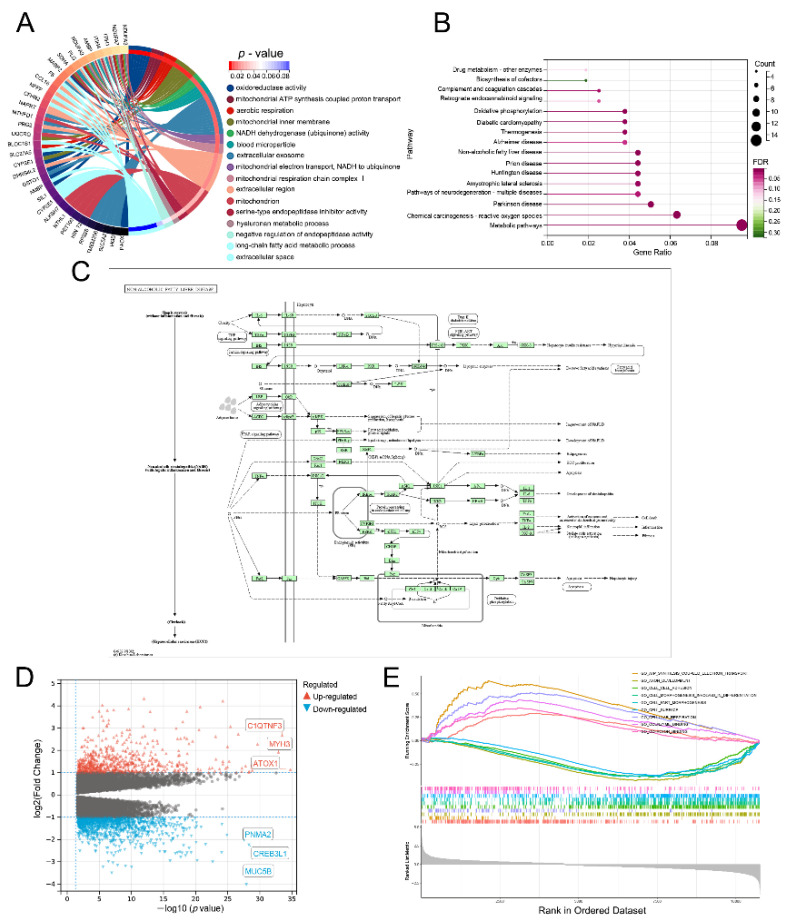
The signaling pathways and biological processes of *lnc-RAB11B-AS1* (**A**) GO analysis. (**B**) Pathway enrichment analysis based on KEGG. (**C**) KEGG pathway: NAFAD. (**D**) Volcano plot of DEGs. Blue and red dots represent the significantly down-regulated and up-regulated DEGs, respectively. (**E**) Ten significantly enriched pathways based on GSEA.

**Figure 5 cells-11-03517-f005:**
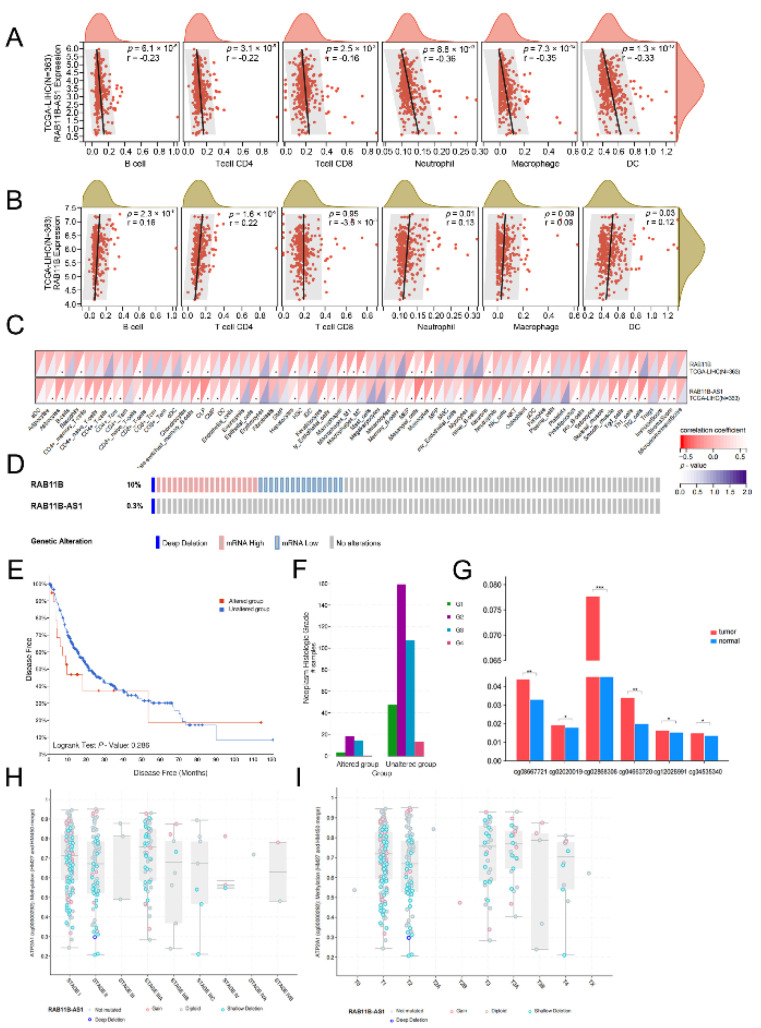
Copy number variations (CNVs) of *RAB11B* in HCC tissues (**A**,**B**) Correlation analysis between *lnc-RAB11B-AS1*, *RAB11B*, and six immune cells in TIMER. (**C**) The relationship between the expression levels of two genes and 64 immune cells using xCell. (**D**) OncoPrint plot of *lnc-RAB11B-AS1* and *RAB11B* alterations. (**E**) Genetic alteration in *RAB11B* might be associated with shorter OS of HCC patients. (**F**) Relationship between the gene mutation level of *RAB11B* and tumor histological grade. (**G**) Methylation levels of the promoter region of *lnc-RAB11B-AS1* were higher in the tumor group compared to the paired normal group. (**H**,**I**) Methylation levels in different clinical stages and T stage. * *p* < 0.05, ** *p* < 0.01, *** *p* < 0.001.

**Figure 6 cells-11-03517-f006:**
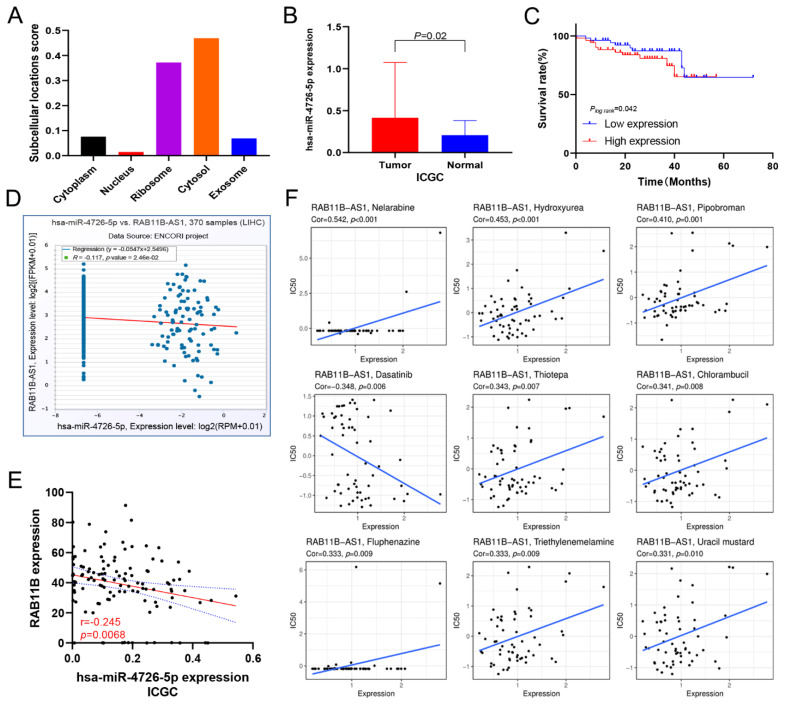
Prediction of potential candidate miRNAs and drug sensitivity analysis (**A**) The prediction of the subcellular localization of *lnc-RAB11B-AS1*. (**B**) The expression status of *hsa-miR-4726-5p* in HCC tissues and normal tissues from ICGC. (**C**) Kaplan-Meier overall survival was analyzed according to *hsa-miR-4726-5p* expression in ICGC cohort. (**D**) The relationship between *lnc-RAB11B-AS1* and *hsa-miR-4726-5p* expression in ENCORI database. (**E**) Correlation analysis of the expression of *RAB11B* and *hsa-miR-4726-5p* in 232 pairs of HCC tissues. (**F**) Relationship between *lnc-RAB11B-AS1* and chemosensitivity to nine drugs.

**Table 1 cells-11-03517-t001:** Relationship between clinicopathological characteristics and expression of *lnc-RAB11B-AS1*.

Characteristics	*n*	*Lnc-RAB11B-AS1* Expression		χ^2^	*p*-Value
		High (n = 184)	Low (n = 184)		
Sex				9.908	**0.002**
Male	248	139 (56.1)	109 (43.9)		
Female	120	46 (38.3)	74 (61.7)		
Age				2.297	0.130
≤60	179	82 (45.8)	97 (54.2)		
>60	188	101 (53.7)	87 (46.3)		
Missing	1	1 (100.0)	0 (0)		
Race				0.603	0.740
Asian	158	75 (47.5)	83 (52.5)		
White	182	89 (48.9)	93 (51.1)		
Others	28	14 (50.0)	14 (50.0)		
Historical risk factors				4.225	0.238
History of hepatitis B or hepatitis C	101	53 (52.5)	48 (47.5)		
Alcohol consumption	65	25 (38.5)	40 (61.5)		
Smoking	8	4 (50.0)	4 (50.0)		
Others	194	102 (52.6)	92 (47.4)		
Clinical stage				4.054	**0.044**
I, II	257	140 (54.5)	117 (45.5)		
III, IV	88	37 (42.0)	51 (58.0)		
Missing	23	7 (30.4)	16 (69.6)		
T				2.472	0.116
T1, T2	275	144 (52.4)	131 (47.6)		
T3, T4	91	39 (42.9)	52 (57.1)		
Missing	2	1 (50.0)	1 (50.0)		
N				4.605	**0.032**
N0	250	121 (48.2)	129 (51.8)		
N1	5	0 (0)	5 (100.0)		
Missing	113	63 (55.8)	50 (44.2)		
M				1.03	0.310
M0	265	134 (50.6)	131 (49.4)		
M1	4	1 (25.0)	3 (75.0)		
Missing	99	49 (49.5)	50 (50.5)		
Histologic grade				4.416	**0.036**
G1, G2	232	125 (53.9)	107 (46.1)		
G3, G4	132	56 (42.4)	76 (57.6)		
Missing	4	3 (75.0)	1 (25.0)		
AFP (μg/L)				8.066	**0.018**
≤20	147	86 (58.5)	61 (41.5)		
20 < AFP ≤ 400	65	28 (43.1)	37 (56.9)		
>400	65	26 (40.0)	39 (60.0)		
Missing	91	44 (48.4)	47 (51.6)		
T-Bil (μmol/L)					
Medical reference value	4	4 (100.0)	0 (0)	4.500	**0.034**
Abnormal value	289	151 (52.2)	138 (47.8)		
Missing	75	29 (38.7)	46 (61.3)		
ALB (g/L)				3.713	0.054
Medical reference value	4	4 (100.0)	0 (0)		
Abnormal value	291	150 (51.5)	141 (48.5)		
Missing	73	30 (41.1)	43 (58.9)		
Creatinine (μmol/L)				0.165	0.685
Medical reference value	182	97 (53.3)	85 (46.7)		
Abnormal value	114	58 (50.9)	56 (49.1)		
Missing	72	29 (40.3)	43 (59.7)		
Child pugh grade				0.018	0.895
A	216	121 (56.0)	95 (44.0)		
B,C	22	12 (54.5)	10 (45.5)		
Missing	130	51 (39.2)	79 (60.8)		
Treatment or therapy				1.703	0.192
Yes	39	16 (41.0)	23 (59.0)		
No	307	160 (52.1)	147 (47.9)		
Missing	22	8 (36.4)	14 (63.6)		
Treatment type				0.003	0.959
Pharmaceutical Therapy	185	93 (50.1)	92 (49.9)		
Radiation Therapy	183	92 (50.3)	91 (49.7)		
Cancer first-degree relatives				2.247	0.134
≤1	65	42 (64.6)	23 (35.4)		
>1	53	27 (50.9)	26 (49.1)		
Missing	250	115 (46.0)	135 (54.0)		
BMI	335	168 (50.1)	167 (49.9)	0.884	0.377
Missing	33	16 (48.5)	17 (51.5)		

*p*-value: Statistically significant results (in bold); TNM: Tumor-node-metastasis; BMI: Body mass index.

**Table 2 cells-11-03517-t002:** Relationship between clinicopathological characteristics and expression of RAB11B.

Characteristics	*n*	*RAB11B* Expression		χ^2^	*p*-Value
		High (n = 184)	Low (n = 184)		
Sex				0.012	0.911
Male	249	124 (49.8)	125 (50.2)		
Female	119	60 (50.4)	59 (49.6)		
Age				0.067	0.796
≤60	176	89 (50.6)	87 (49.4)		
>60	191	94 (49.2)	97 (50.8)		
Missing	1	1 (100.0)	0 (0)		
Race				0.854	0.653
Asian	158	75 (47.5)	83 (52.5)		
White	181	93 (51.4)	88 (48.6)		
Others	29	16 (55.2)	13 (44.8)		
Historical risk factors				2.200	0.532
History of hepatitis B or hepatitis C	101	51 (50.5)	50 (49.5)		
Alcohol consumption	65	28 (43.1)	37 (56.9)		
Smoking	9	4 (44.4)	5 (55.6)		
Others	193	103 (53.4)	90 (46.6)		
Clinical stage				3.426	0.064
I, II	256	134 (52.3)	122 (47.7)		
III, IV	89	36 (40.4)	53 (59.6)		
Missing	23	14 (60.9)	9 (39.1)		
T				2.472	0.116
T1, T2	275	144 (52.4)	131 (47.6)		
T3, T4	91	39 (42.9)	52 (57.1)		
Missing	2	1 (50.0)	1 (50.0)		
N				0.850	0.357
N0	251	121 (48.2)	130 (51.8)		
N1	4	1 (25.0)	3 (75.0)		
Missing	113	62 (54.9)	51 (45.1)		
M				0.025	0.875
M0	265	122 (46.1)	143 (53.9)		
M1	4	2 (50.0)	2 (50.0)		
Missing	99	60 (60.6)	39 (39.4)		
Histologic grade				5.920	**0.015**
G1, G2	233	127 (54.3)	106 (45.7)		
G3, G4	131	54 (41.2)	77 (58.8)		
Missing	4	3 (75.0)	1 (25.0)		
AFP (μg/L)				0.581	0.748
≤20	147	72 (49.0)	75 (51.0)		
20<AFP ≤ 400	66	32 (48.5)	34 (51.5)		
>400	64	34 (53.1)	30 (46.9)		
Missing	91	46 (50.5)	45 (49.5)		
T-Bil (μmol/L)				0.894	0.344
Medical reference value	4	3 (75.0)	1 (25.0)		
Abnormal value	289	148 (51.2)	141 (48.8)		
Missing	75	33 (44.0)	42 (56.0)		
ALB (g/L)				0.920	0.337
Medical reference value	4	3 (75.0)	1 (25.0)		
Abnormal value	291	148 (50.9)	143 (49.1)		
Missing	73	33 (45.2)	40 (54.8)		
**Creatinine (μmol/L)**				0.109	0.741
Medical reference value	182	89 (48.9)	93 (51.1)		
Abnormal value	114	58 (50.9)	56 (49.1)		
Missing	72	37 (51.4)	35 (48.6)		
Child pugh grade				0.092	0.762
A	217	111 (51.2)	106 (48.8)		
B,C	22	12 (54.5)	10 (45.5)		
Missing	129	61 (47.3)	68 (52.7)		
Treatment or therapy				0.205	0.651
Yes	40	21 (52.5)	19 (47.5)		
No	306	149 (48.7)	157 (51.3)		
Missing	22	14 (63.6)	8 (36.4)		
Treatment type				0.330	0.566
Pharmaceutical Therapy	185	95 (51.4)	90 (48.6)		
Radiation Therapy	183	88 (48.1)	94 (51.9)		
Cancer first-degree relatives				1.238	0.266
≤1	66	36 (54.5)	30 (45.5)		
>1	52	23 (44.2)	29 (55.8)		
Missing	250	125 (50.0)	125 (50.0)		
BMI	335	166 (49.6)	169 (50.4)	1.112	0.267
Missing	33	18 (54.5)	15 (45.5)		

**Table 3 cells-11-03517-t003:** Univariate and multivariate Cox regression analyses of clinicopathologic characteristics associated with OS in TCGA samples.

Variable		Univariable			Multivariable		
		HR	95%CI	*p*	HR	95%CI	*p*
Gender		1.119	0.771–1.624	0.556	1.192	0.610–2.330	0.725
Age		1.013	0.998–1.027	0.083	1.023	0.998–1.051	0.076
BMI		1.019	0.967–1.028	0.819	1.031	0.979–1.086	0.377
Race	1	Reference			Reference		
	2	1.301	0.879–1.925	0.189	0.539	0.255–1.141	0.106
	3	1.526	0.647–3.599	0.334	1.812	0.232–14.105	0.570
Clinical stage		1.312	1.011–1.703	0.041	2.628	1.599–4.321	<0.001
T		2.562	1.770–3.707	<0.001	20.208	0.869–469.5	0.061
N		1.991	1.487–8.144	0.038	8.846	1.013–1.096	0.013
M		3.907	1.225–12.47	0.021	1.233	0.135–11.281	0.853
Histologic grade		1.060	0.726–1.547	0.762	1.119	0.631–1.983	0.701
AFP (μg/L)	1	Reference			Reference		
	2	1.352	0.621–2.941	0.447	1.102	0.321–2.564	0.854
	3	0.924	0.385–2.220	0.860	1.224	0.278–2.402	0.714
T-Bil (μmol/L)		1.356	0.185–9.932	0.764	1.356	0.185–9.932	0.764
ALB (g/L)		1.398	0.101–10.240	0.742	1.284	0.175–9.434	0.806
Creatinine (μmol/L)		1.710	1.067–1.374	0.031	1.744	1.297–1.862	0.027
Child pugh grade		2.205	0.296–16.401	0.365	1.993	0.236–16.846	0.526
Treatment type		1.231	0.858–1.766	0.260	1.734	0.979–3.071	0.059
Treatment or therapy		1.039	0.592–1.822	0.895	1.176	0.492–2.807	0.716
*Lnc-RAB11B-AS1*		0.814	0.696–0.951	0.009	0.799	0.656–0.972	0.025
*RAB11B*		0.651	0.467–0.909	0.012	0.898	0.978–1.000	0.041

**Table 4 cells-11-03517-t004:** Relationship between clinicopathological characteristics and expression of *RAB11B* protein in HCC patients.

Characteristics	*n*	*RAB11B* Expression		χ^2^	*p*-Value
		High (n = 25)	Low (n = 65)		
Sex				0.339	0.560
Male	80	23 (28.6)	57 (71.4)		
Female	10	2 (20.0)	8 (80.0)		
Age				0.543	0.461
≤60	71	21 (29.6)	50 (70.4)		
>60	19	4 (21.1)	15 (78.9)		
Pathology grade				15.691	<0.001
I	3	2 (66.7)	1 (33.3)		
II	43	19 (44.2)	24 (55.8)		
III	44	4 (9.1)	40 (90.9)		
Clinical stage				2.018	0.365
1	63	20 (31.7)	43 (68.3)		
2	25	5 (20.0)	20 (80.0)		
3	2	0 (0)	2 (100.0)		
T				2.225	0.329
T1	63	20 (31.7)	43 (68.3)		
T2	24	5 (20.8)	19 (79.2)		
T3	3	0 (0)	3 (100.0)		
Recurrence				1.522	0.217
Yes	49	11 (22.4)	38 (77.6)		
No	41	14 (34.1)	27 (65.9)		
HBsAg				2.811	0.094
Positive	70	16 (22.9)	54 (77.1)		
Negative	19	8 (42.1)	11(57.9)		
Unknown	1	1 (100.0)	0 (0)		
HBcAb				0.018	0.894
Positive	80	21 (26.3)	59 (73.7)		
Negative	7	2 (28.6)	5 (71.4)		
Unknown	3	2 (66.7)	1 (33.3)		
AntiHCV				2.815	0.093
Positive	1	1 (100.0)	0 (0)		
Negative	86	22 (25.6)	64 (74.4)		
Unknown	3	2 (66.7)	1 (33.3)		
T-Bil (μmol/L)				4.107	0.043
Medical reference value	63	21 (33.3)	42 (66.7)		
Abnormal value	25	3 (12.0)	22 (88.0)		
Unknown	2	1 (50.0)	1 (50.0)		
ALT (U/L)				0.224	0.636
Medical reference value	52	15 (28.8)	37 (71.2)		
Abnormal value	37	9 (24.3)	28 (75.7)		
Unknown	1	1 (100.0)	0 (0)		
AFP (μg/L)				0.716	0.397
≤20	36	10 (27.8)	26 (72.2)		
>20	53	14 (26.4)	39 (73.6)		
Unknown	1	1 (100.0)	0 (0)		
GGT(U/L)				3.903	0.048
≤40	30	12 (40.0)	18 (60.0)		
>40	59	12 (20.3)	47 (79.7)		
Unknown	1	1 (100.0)	0 (0)		
PD-L1 expression				9.357	0.002
Low	39	18 (46.2)	21 (53.8)		
High	45	7 (15.6)	38 (84.4)		
Unknown	6	0 (0)	6 (100.0)		
CTLA4 expression				0.786	0.375
Low	2	0 (0)	2 (100.0)		
High	81	23 (28.4)	58 (71.6)		
Unknown	7	2 (28.6)	5 (71.4)		

HBsAg: Hepatitis B surface antigen; HBcAb: Hepatitis B core antibody; AntiHCV: Antibody to Hepatitis surface antigen; TB: Total bilirubin; AFP: Alpha-fetoprotein; GGT: Gamma-glutamyl transferase; PD-L1: Programmed cell death ligand-1; CTLA4: Cytotoxic T lymphocyte-associated protein 4. The medical reference value of T-Bil (μmol/L) and ALT (U/L) is 5.13~22.24, 7~40, respectively.

## Data Availability

The data that support the findings of this study are openly available in online databases mentioned in the manuscript, and also available from the corresponding author upon reasonable request.

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
