# Peer review of "Bioinformatics Analysis and Validation of the Role of Lnc-RAB11B-AS1 in the Development and Prognosis of Hepatocellular Carcinoma"

_cells, 2022, doi:10.3390/cells11213517_

Round 1
Reviewer 1 Report
The present Article, cells-1971685, entitled: (Bioinformatics analysis and validation of the development and prognosis role of lnc-RAB11B-AS1 in hepatocellular carcinoma). The Research article is informative and written in a concise way. However, I have the following comments:
- Authors have done an excellent bioinformatics analysis, and they made Immunohistochemical staining for tissues (figure 3A) however I can not see ethical approval to perform and represent the results related to tissue staining????
- There are many figure labels with very poor quality ( can not be seen even at magnification 400 %) e.g : (figure 3E, F), (figure 4 A to E), (figure 5c) and (figure 6d)
Some minor comments:
- P- value written in many cases in italic style and many cases written in regular style, so check all of them and change them to italic
- In page 14 line 363 CD+T need to change to superscript format.
Author Response
Response to Reviewer 1
[General Comment]
The present Article, cells-1971685, entitled: (Bioinformatics analysis and validation of the development and prognosis role of lnc-RAB11B-AS1 in hepatocellular carcinoma). The Research article is informative and written in a concise way. However, I have the following comments.
Response: Thank you for your comments. We have gone through your comments carefully and tried our best to address them one by one. We hope the manuscript has been improved accordingly.
[Comment 1] Authors have done an excellent bioinformatics analysis, and they made Immunohistochemical staining for tissues (figure 3A) however I can not see ethical approval to perform and represent the results related to tissue staining????
Response: Thanks for your nice reminder. The approval code of the ethical committee is SHYJS-CP-1901001, and approved by the ethics committee of Shanghai Outdo Biotech Co., Ltd. The ethical approval document has been attached to the editor before. We revised the sentence in subsection 2.2 as follows: “The protocol was approved by the ethics committee of Shanghai Outdo Biotech Co., Ltd.”
[Comment 2] There are many figure labels with very poor quality (can not be seen even at magnification 400 %) e.g: (figure 3E, F), (figure 4 A to E), (figure 5c) and (figure 6d)
Response: Thank you very much. We revised most of the figure captions to make them clearer.
Some minor revisions for the authors to consider:
[Minor Comment 1] P- value written in many cases in italic style and many cases written in regular style, so check all of them and change them to italic.
Response: Thank you very much. Revised accordingly.
[Minor Comment 2] In page 14 line 363: CD+T need to change to superscript format.
Response: Thanks for your kind reminders. Revised accordingly.
Reviewer 2 Report
The current report is a promising addition to the current understanding and enhancement of HCC pathogenesis. The current explicit, ensemble bioinformatics analysis ranging from IHC expressions in HCC to time-dependent analysis mainly revealed that low expressed lnc-RAB11B-AS1/RAB11B is a hazard factor in HCC prognostication. Survival analysis as validation demonstrated that patients with higher RAB11B expression showed longer OS and DFS, suggesting that RAB11B might also be a protective factor in HCC development. The localization and causality were both clearly addressed. Concerns:
-Numerous grammatical errors were easily spotted by a person speaking ESL. For the reputation, this paper must be extensively revised paragraph by paragraph again by a native English speaker. Thank you.
-Several examples of the numerous errors or unclear text are as follows:
Line 38: people?
Lines 107–109: A total of 90 HCC patients with cancer tissues and paired non-cancerous tissues were enrolled in our study, which were all histologically confirmed cases.?
Lines 116, 117: and we imported the gene name to get a network diagram containing 20 proteins interacted with RAB11B.?
Lines 242, 243: the results in GSE144269 (Figure 2G, 2H) uncovered the same trend between two genes and the prognosis, (unclear-- which two genes?)
(Similarly, lines 467–470?)
…
-“HCC” addressed throughout the entire report sounds highly unclear to hepatologists. Please specify the status of “HCC treatment” for each data source-- e.g., de novo? treated? recurrent? Consult hepatologists. Please address something if not feasible for certain databases. Address some concise discussions on the treatment status currently lacking in the models.
-Try addressing more discussions (concisely) on the opposite-- in Abstract, “oncogene” was obviously mentioned.
Author Response
Response to Reviewer 2
[General Comment] The current report is a promising addition to the current understanding and enhancement of HCC pathogenesis. The current explicit, ensemble bioinformatics analysis ranging from IHC expressions in HCC to time-dependent analysis mainly revealed that low expressed lnc-RAB11B-AS1/RAB11B is a hazard factor in HCC prognostication. Survival analysis as validation demonstrated that patients with higher RAB11B expression showed longer OS and DFS, suggesting that RAB11B might also be a protective factor in HCC development. The localization and causality were both clearly addressed.
Response: Thank you very much for agreeing with us on the intention of this manuscript. We have read your comments carefully and tried our best to address them one by one, especially in terms of providing a discussion on the oncogenic role of RAB11B-AS1 in other tumors and the treatment status currently lacking in the models. We hope that the manuscript has been improved towards your standards after this revision.
[Comment 1] Numerous grammatical errors were easily spotted by a person speaking ESL. For the reputation, this paper must be extensively revised paragraph by paragraph again by a native English speaker. Thank you.
Response: Thank you for your suggestions. According to your advice, this manuscript was edited for proper English language, grammar, punctuation, spelling, and overall style by one or more of the highly qualified native English-speaking editors at Edanz (https://www.edanz.com/ac; We thank H. Nikki March, PhD, from Edanz for editing a draft of our manuscript). Edanz specializes in editing and proofreading scientific manuscripts for submission to peer-reviewed journals.
[Comment 2] Several examples of the numerous errors or unclear text are as follows:
- Line 38: people?
- Lines 107–109: A total of 90 HCC patients with cancer tissues and paired non-cancerous tissues were enrolled in our study, which were all histologically confirmed cases.?
- Lines 116, 117: and we imported the gene name to get a network diagram containing 20 proteins interacted with RAB11B.?
- Lines 242, 243: the results in GSE144269 (Figure 2G, 2H) uncovered the same trend between two genes and the prognosis, (unclear-- which two genes?)
- (Similarly, lines 467–470?)
Response: Thank you very much for pointing this out. We revised the sentence as follows:
- Pg1, Ln37-38: HCC has become the fourth most common malignant tumor and the second most lethal cause of cancer in China, which seriously increases the disease burden.
- Pg3, Ln113-115: A total of 90 histologically confirmed HCC cases were enrolled in our study, which contain 90 pairs of cancer tissues and adjacent non-cancerous tissues.
- Pg3, Ln123-125: The default species “Homo sapiens” was selected, and we constructed PPI networks by GeneMANIA online to analyze the interaction between RAB11B and other functional proteins.
- Pg9, Ln255-257: As an independent validation dataset, the results of RAB11B-AS1 and RAB11B in GSE144269 uncovered the similar prognostic trend (Figure 2G, 2H). HCC patients with decreased expression of the above two genes experience shorter survival time but no statistical significance was observed (P=0.12, P=0.21, respectively) due to the deficiency of samples in GEO data (n=68).
- Pg20, Ln499-500: A highly positive relationship was observed between the expression of RAB11B-AS1 and RAB11B.
[Comment 3]
“HCC” addressed throughout the entire report sounds highly unclear to hepatologists. Please specify the status of “HCC treatment” for each data source-- e.g., de novo? treated? recurrent? Consult hepatologists. Please address something if not feasible for certain databases. Address some concise discussions on the treatment status currently lacking in the models.
Response: Thank you for your comment. The datasets in our article are derived from TCGA, ICGC and GEO. Firstly, TCGA was designed primarily for molecular studies, initial cases were selected from multiple institutions with suitable banked tissues, most were untreated primary cases, thus these cases do not constitute a consecutive series. We added the status of “HCC treatment” for “TCGA-LIHC” and “ICGC-LIRI-JP” datasets by consulting relevant professionals and carefully reviewing the clinical data of HCC patients. For the two GEO datasets, GSE84402 and GSE144269, both sequencing data were uploaded by the investigators, but no systematic clinical information was collected, so we can not know the specific treatment status. Previous research concluded that visualization techniques and descriptive indicators are effective in assessing large databases and could give researchers and clinicians a clearer understanding of how survival differs by disease subtype, treatment status, and practice patterns, thus we also compared the median survival time of HCC patients with different treatment status in our article.
We added the relevant contents to our manuscript as follows:
Pg3, Ln99-102: Among 347 HCC patients with known treatment statuses, 239 patients did not receive treatment (4 of them had another new primary tumor), 18 patients were treated, and the remaining 90 cases were recurrent (13 of them had relapses of other tumors, such as lung and bone cancer).
Pg20, Ln501-517: According to the former study, visualization techniques and descriptive indicators can be used to reflect how survival differs by disease subtype, treatment status, and treatment practice patterns[42]. Through screening the clinicopathological information in the database and referring to relevant literature, we first differentiated HCC patients with different treatment statuses in TCGA, among which 239 were untreated, 18 were treated, 90 were recurrent, and 22 were unreported cases. Subsequently, median survival differences between patients with the above three known statuses were assessed. We found that the overall median survival times of untreated, treated, and recurrent HCC patients were 1.43 (IQR 0.05–2.01) years, 1.48 (IQR 0.10–2.43) years, and 2.43 (IQR 1.02–3.48) years, respectively. While in the ICGC database, we only obtained information on whether these HCC patients were treated and the corresponding treatment patterns (177 were untreated and 63 were treated with chemotherapy, surgery or other trerapies). We found that the overall median survival of untreated patients was 2.05 years (IQR 1.39–3.21) and that of treated patients was 2.31 years (IQR 1.56–2.88). Patients diagnosed with recurrent HCC experienced longer survival times than treated HCC patients, consistent with our hypothesis. Unfortunately, for the GSE144269 and GSE84402 datasets, even after reviewing the original text carefully, we have no information on the treatment status of HCC patients.
References:
Gilbert A, Williams C, Azuero A, Burkard ME, Kenzik K, Garrett-Mayer E, et al. Utilizing Data Visualization to Identify Survival and Treatment Differences Between Women With De Novo and Recurrent Metastatic Breast Cancer. Clinical breast cancer 2021; 21(4):292-301. (This reference is from revised manuscripts)
Liu J, Lichtenberg T, Hoadley KA, Poisson LM, Lazar AJ, Cherniack AD, et al. An Integrated TCGA Pan-Cancer Clinical Data Resource to Drive High-Quality Survival Outcome Analytics. Cell 2018; 173(2):400-416.e411.
[Comment 4]
Try addressing more discussions (concisely) on the opposite-- in Abstract, “oncogene” was obviously mentioned.
Response: Thanks for your comment. Through consulting literature materials, we have supplemented the discussion with the carcinogenic role of RAB11B-AS1 in other tumors.
Pg20, Ln476-491: RAB11B-AS1 is a member of the RAS oncogene family and it was found to be dysregulated in several tumors, affecting their biological progress and even resistance to therapeutic agents[38]. As an oncogene, Niu and colleagues demonstrated that RAB11B-AS1 stimulated the development and growth of breast cancer cells in vitro, as well as tumor angiogenesis and distant metastasis of breast cancer, without affecting primary tumor growth in mice[39]. Li and colleagues found that RAB11B-AS1 is overexpressed in lung cancer tissues and associated with poor prognosis in lung cancer[13]. Feng et al. detected the expression of RAB11B-AS1 in 83 gastric cancer tissues and normal tissues using qRT-PCR and found that upregulated RAB11B-AS1 in gastric cancer tissues was related to higher stage, lymph node metastasis, distant metastasis, and degree of tumor differentiation. Mechanistically, RAB11B-AS1 may inhibit the proliferation, migration, and invasion of gastric carcinoma cells by up-regulating miR-628-3p and increasing the sensitivity to cisplatin[40].
References: (The references are from revised manuscripts)
[13] Li T, Wu D, Liu Q, Wang D, Chen J, Zhao H, et al. Upregulation of long noncoding RNA RAB11B-AS1 promotes tumor metastasis and predicts poor prognosis in lung cancer. Annals of translational medicine 2020; 8(9):582.
[38] Rothzerg E, Ho XD, Xu J, Wood D, Märtson A, Kõks S. Upregulation of 15 Antisense Long Non-Coding RNAs in Osteosarcoma. Genes 2021; 12(8).
[39] Niu Y, Bao L, Chen Y, Wang C, Luo M, Zhang B, et al. HIF2-Induced Long Noncoding RNA RAB11B-AS1 Promotes Hypoxia-Mediated Angiogenesis and Breast Cancer Metastasis. Cancer research 2020; 80(5):964-975.
[40] Yang N B, Jia X D, Kou Y. lncRNA RAB11B-AS1 up-regulates miR-628-3p and inhibits proliferation and migration of human gastric cancer cell lines. Journal of Basic Medicine and Clinical Medicine 2021; 41 (12): 1762-1766.
Thanks again for your valuable and thoughtful comments. We hereby resubmit the revised manuscript and hope that all corrections are satisfactory. Please feel free to contactus with any questions and we look forward to your decision.
Round 2
Reviewer 1 Report
The present Article, cells-1971685R1, entitled: (Bioinformatics analysis and validation of the development and prognosis role of lnc-RAB11B-AS1 in hepatocellular carcinoma)
The Research article is informative and written in a concise way and i can see that authors have addressed all comments and manuscript is accepted in the current form for publication.
Reviewer 2 Report
A few typographical errors are still found in the 4 tables and 6 figures; e.g., in Table 1, the greater-than sign: >>?; History hepatocarcinoma risk factors? Hepatitis B、C?. Please have a native English speaker revise the tables (heading, entry, content, footnote, etc.) and figures (footnote) again and finalize the current manuscript! Thank you.